# Modelling the modulation of cortical Up-Down state switching by astrocytes

**Lisa Blum Moyse**[1,2], **Hugues Berry**[1,2]*

**1** Inria, Villeurbanne, France, **2** LIRIS UMR5205, University of Lyon, Villeurbanne, France

* hugues.berry@inria.fr

## Abstract

Up-Down synchronization in neuronal networks refers to spontaneous switches between periods of high collective firing activity (Up state) and periods of silence (Down state). Recent experimental reports have shown that astrocytes can control the emergence of such Up-Down regimes in neural networks, although the molecular or cellular mechanisms that are involved are still uncertain. Here we propose neural network models made of three populations of cells: excitatory neurons, inhibitory neurons and astrocytes, interconnected by synaptic and gliotransmission events, to explore how astrocytes can control this phenomenon. The presence of astrocytes in the models is indeed observed to promote the emergence of Up-Down regimes with realistic characteristics. Our models show that the difference of signalling timescales between astrocytes and neurons (seconds versus milliseconds) can induce a regime where the frequency of gliotransmission events released by the astrocytes does not synchronize with the Up and Down phases of the neurons, but remains essentially stable. However, these gliotransmission events are found to change the localization of the bifurcations in the parameter space so that with the addition of astrocytes, the network enters a bistability region of the dynamics that corresponds to Up-Down synchronization. Taken together, our work provides a theoretical framework to test scenarios and hypotheses on the modulation of Up-Down dynamics by gliotransmission from astrocytes.

## Author summary

Neural networks in many brain regions can display synchronized activities. During the so-called "Up-Down" synchronization regimes for instance, the whole local population of neurons switches in a spontaneous and synchronized fashion between phases of high activity (Up states) and phases of low activity (Down states). The mechanisms responsible for this behaviour are still not well understood, but recent experimental reports have suggested that another type of brain cells, the astrocytes, at least partly control these oscillations. Astrocytes are increasingly believed to play a role in the propagation of signals between neurons, via their connections to neuronal synapses, but how this mechanism could control Up-Down regimes is not understood. To address this issue we present here simple mathematical models of neuronal networks that incorporate astrocytes in addition

**Data Availability Statement:** All the code used to generate the data of this article is freely available online on ModelDB: http://modeldb.yale.edu/267310, with access code: AstroUpDownRO22. This access code will be removed when the manuscript is accepted for publication.

**Funding:** The author(s) received no specific funding for this work.

**Competing interests:** The authors have declared that no competing interests exist.

to neurons according to various levels of description. Using bifurcation analysis and numerical simulations we explore how astrocytes control Up-Down synchronization of the neuronal networks. In particular, astrocytes in the model are found to change the localization of the bifurcation points in the parameter space, so that the neurons enter the region of Up-Down regime when astrocytes are present. We also give some theoretical predictions that can be tested experimentally to test the validity of our models.

## Introduction

Collective behaviors are characterized by the emergence of a coherent group behavior on the basis of simple interactions between the individuals of the group. Understanding the relationship between the properties of the individuals and the coordinated behavior at the population level usually demands theoretical approaches, for instance from theoretical physics [1–3]. Among the numerous forms of collective behaviors reported in the brain, Up-Down dynamics are characterized by spontaneous switches between periods of intense firing of the whole neuronal population (Up state) and periods of silence (Down state), even in the absence of external inputs [4–7].

The cellular and network mechanisms at the origin of cortical Up-Down dynamics are still not well understood. For a large part, the phenomenon seems intrinsic to the cortical networks since it has been observed in cortical slices [6] and survives *in vivo* when the connections between cortex and thalamus, its main source of inputs, are lesioned [4]. A number of theoretical studies have proposed intrinsic mechanisms to explain cortical Up-Down dynamics, i.e., mechanisms that originate from the neurons themselves [8–11]. These proposals usually postulate some sort of activity-dependent negative feedback of the firing rate, according to which individual neurons tend to decrease their firing rate after sustained periods of firing, and to increase it after sustained periods of silence. In the simplest cases, this negative feedback can rely on a slow adaptation current [9, 12–14] or short term plasticity [10, 11], for instance.

However, the existence of a rhythm generation mechanism intrinsic to the neurons of the network does not mean that the input from other brain regions cannot play a role. Several experimental studies have evidenced that oscillatory inputs from the thalamus do strongly impact or even trigger cortical Up-Down dynamics [15–17]. In agreement with these observations, several theoretical studies have been proposed to study Up-Down dynamics in the framework of the interplay between an intrinsic activity-dependent negative feedback of the firing rate and an external input to the network [18–20].

Recently, astrocytes have been identified by experimental studies as a new potential actor of population oscillations in the brain [21–23]. Astrocytes are non-neuronal neural cells that, together with oligodendrocytes, ependymal cells and microglia form the glial cells [24, 25]. Astrocytes can ensheath synaptic elements, thus forming a "tripartite" synapse where signalling information can flow between the presynaptic neuron, the postsynaptic neuron and the astrocyte [26, 27]. Indeed, at the tripartite synapse, astrocytes integrate neuronal activity as a complex transient signal of their intracellular $Ca^{2+}$ concentration [28, 29]. In addition, astrocytic intracellular $Ca^{2+}$ signals can, at least under certain conditions, trigger the release by the astrocyte of neuroactive molecules called gliotransmitters that may in turn modulate neuronal information transfer [30, 31]. The existence in physiological conditions of such a bilateral signalling between neurons and astrocytes is still debated among experimental neuroscientists, in particular regarding the impact of gliotransmitters on neurons [32, 33]. But if confirmed, it

could explain the accumulated experimental evidence of the implication of astrocytes in information treatment in the brain [30, 34, 35].

Recently, a series of experimental studies has suggested that astrocytes are another intrinsic mechanism for the generation of Up-Down dynamics in cortical networks [36, 37]. In cortical slices, they have observed that increasing the calcium activity of a single astrocyte is enough to roughly double the probability to observe an Up state in the surrounding neurons, with no change of the amplitude nor the duration of these Up states [36]. In vivo experiments further showed that increasing calcium activity in a local population of astrocytes was temporally correlated to a shift of the local population of neurons to the Up-Down regime [37].

In spite of these significant experimental observations, the mechanism by which astrocytes modulate Up-Down cortical dynamics is still unknown. In particular, it is not understood how the modulation by astrocytes interact or rely on the other identified mechanisms of Up-Down state generation in the neurons. Here, we propose a mathematical model to explore the possible mechanisms by which astrocytes control the emergence of Up-Down dynamics of their surrounding neuronal network populations. To that aim, we extended the model proposed by [20] for Up-Down dynamics in a network of excitatory and inhibitory cortical neurons with a population of astrocytes. This provided us with theoretical tools to understand how the release of gliotransmitters by the astrocytes alters the dynamics of neuronal network towards the emergence of Up-Down dynamics.

## Materials and methods

### Rate model

The model of [20] was designed to study the emergence of Up-Down dynamics in a neural network composed of an excitatory population E connected to an inhibitory population I in a all-to-all manner (Fig 1). The excitatory population is endowed with an adaptation mechanism $a$, that implements an additive hyperpolarizing current to the population E and that grows with its firing rate. Adaptation is therefore the main intrinsic mechanism for the emergence of Up-Down dynamics in the model. However, each population also receives a fluctuating external input. The firing rate dynamics of this model is given by

$$\tau_E \frac{dr_E}{dt} = -r_E(t) + \phi_E(I_E - a(t) + \sigma \xi_E(t)) \tag{1}$$

and

$$\tau_I \frac{dr_I}{dt} = -r_I(t) + \phi_I(I_I + \sigma \xi_I(t)) \tag{2}$$

where $r_X$ is the firing rate of population $X = \{E, I\}$, $I_X$ its recurrent inputs (from the populations of the system, see below). Several experimental studies have evidenced that oscillatory inputs coming from the thalamus or other subcortical areas have a strong influence on cortical Up-Down dynamics [15–17]. The external input $\xi_X$ emulates such an oscillatory external input, as an independent Ornstein-Uhlenbeck process (see [20] for details). $\tau_X$ is the time constant of population $X$, and its transfer function:

$$\phi_X(x) = g_X[x - \theta_X]_+ \tag{3}$$

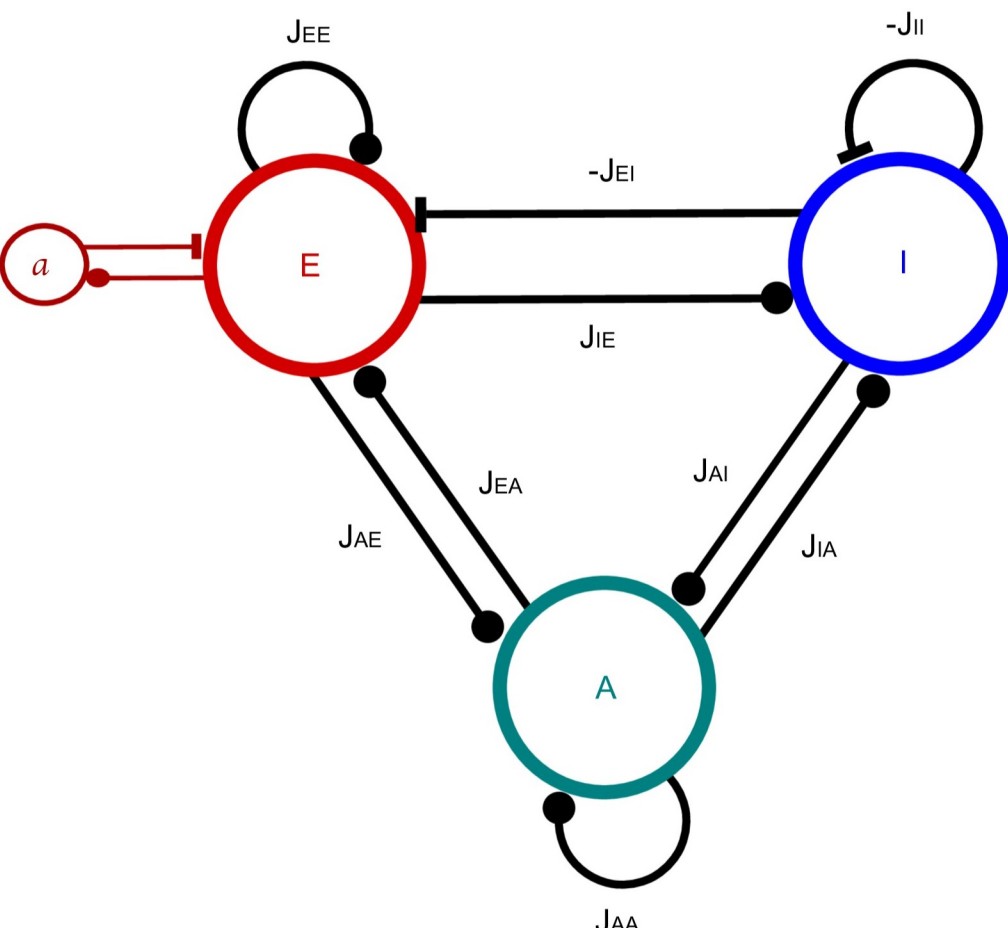

**Fig 1. Interactions between the three populations of the model.** E for excitatory neurons, I for inhibitory neurons and A for astrocytes. *a* represents the adaptation mechanism of E cells. Lines terminated with a full circle represent positive interactions whereas those terminated with a bar represent inhibition (of I cells on E and I, and adaptation *a* on E cells). $J_{XY}$ represent the synaptic strength from the *Y* population toward the *X* population. In case of inhibition these terms are negative.

with rectification $[z]_+ = z$ if $z > 0$ and 0 otherwise. The dynamics of the adaptation current $a(t)$ is given by:

$$\tau_a \frac{da}{dt} = -a(t) + \beta r_E(t) \tag{4}$$

Eqs (1) to (4) define the model proposed in [20]. Here, we extended it to account for the impact of astrocytes on the network.

Astrocytes express a variety of receptors at their membranes, that bind the neurotransmitters or neuromodulators released by the presynaptic elements at the tripartite synapse, including glutamate, GABA, acetylcholine or dopamine [25, 26, 38]. Through these receptors, neuronal activity is integrated inside the astrocyte, which eventually converges in a complex signal of astrocytic intracellular $Ca^{2+}$ [28, 29]. In response to this $Ca^{2+}$ transient, astrocytes can, at least under certain conditions, release in the synapse a variety of molecules, referred to as "gliotransmitters". Upon binding to the pre- or post-synaptic element of the tripartite synapse, gliotransmitters can in turn hyperpolarize or depolarize the neuronal membrane [30–

32]. Interestingly, whereas the astrocytic cytosolic calcium transients are very slow events, especially in the soma (around 10–20 sec on average), gliotransmitter release events are much faster (around 1 sec) [37, 39].

According to this oversimplifying birds-eye view of neuron-astrocyte interactions, the astrocytic response to presynaptic neuronal activity is similar to the process of neuronal integration: presynaptic neuronal activity is integrated in astrocytes as a calcium trace that triggers a peak-like release of gliotransmitters that in turn affects postsynaptic membrane voltage. The major differences are *i*) integration time-scales and gliotransmitter release dynamics in astrocytes are different from electrical signalling in neurons and *ii*) the equivalent of inhibitory/ hyperpolarizing neuronal inputs that decrease the membrane potential does not seem to exist for astrocytic $Ca^{2+}$. To model astrocyte activity, we thus opted for same formalism of rate equations as Eqs (1) or (2), but with different time scales and expressed the rate of gliotransmitter release by the astrocyte as

$$\tau_A \frac{dr_A}{dt} = -r_A(t) + \phi_A(I_A + \sigma\xi_A(t)) \tag{5}$$

with the constraint $\tau_A \gg \tau_I$ and $\tau_A \gg \tau_E$. To our knowledge, there are no a priori reasons to consider that external inputs to the cortical network under study are restricted to a given subtype of brain cells, either excitatory neurons, inhibitory neurons or astrocytes. Therefore, we fed an oscillatory external input to all cell populations, including astrocytes (variable $\xi_A$ above).

Whereas one expects positive values for the firing threshold $\theta_X$ of neurons in Eq (3) (i.e. neurons remain silent below a threshold of their input), we will favor negative values for $\theta_A$, in order to account for the spontaneous calcium activity of astrocytes [25].

We now can give a definition for the three internal recurrent inputs:

$$I_X = J_{XE}r_E(t) + J_{XI}r_I(t) + J_{XA}r_A(t) \tag{6}$$

with now $X = \{E, I, A\}$. The synaptic couplings $J_{XY}$ (with $X, Y = \{E, I, A\}$), describe the strength of the connection from population Y to X. They verify $J_{XE} > 0$ (excitatory), $J_{EI} < 0$, $J_{II} < 0$ (inhibitory) and $J_{AX} \geq 0$ (i.e. both E and I increase the rate of gliotransmitter release in astrocytes).

A fixed-point and linear stability analysis of the rate model defined by Eqs (1) to (6) is provided in S1 Text. The values of the parameters in the equations above are given in Table 1.

## Spiking model

We also modeled the three-population $\{E, I, A\}$ system of Fig 1 by expressing it as a stochastic spiking network model instead of the firing rate framework of Section Materials and methods for the Rate model. Following the same principle as above, where we used a classical neuron rate equation to model astrocyte gliotransmitter release, we used here leaky integrate and fire equations to model both neuronal membrane potential and the release of gliotransmitters by astrocytes. Hence, the membrane potential of the two populations of neurons reads:

$$\tau_E \frac{dV_i^E}{dt} = -(V_i^E - V_L) + I_{rec,i}^E(t) + I_{ext,i}^E(t) - K_a I_{a,i}(t) \tag{7}$$

$$\tau_I \frac{dV_i^I}{dt} = -(V_i^I - V_L) + I_{rec,i}^I(t) + I_{ext,i}^I(t) \tag{8}$$

**Table 1. Parameters used for the rate model Eqs (1) to (6).**

| Parameter | Value | Definition | Parameter | Value | Definition |
|---|---|---|---|---|---|
| $\tau_E$ | 10 ms | time const., E | $\tau_I$ | 2 ms | time const., I |
| $\tau_A$ | 20 ms | time const., A | $\tau_a$ | 500 ms | time const., adaptation |
| $\theta_I$ | 25 | threshold, I | $\theta_A$ | -3.5 | threshold, A |
| $J_{EE}$ | 5 s | strength, E $\rightarrow$ E | $J_{EI}$ | -1 s | strength, I $\rightarrow$ E |
| $J_{II}$ | -0.5 s | strength, I $\rightarrow$ I | $J_{IE}$ | 10 s | strength, E $\rightarrow$ I |
| $J_{AA}$ | 0.1 s | strength, A $\rightarrow$ A | $J_{EA}$ | 1 s | strength, A $\rightarrow$ E |
| $J_{IA}$ | 0.5 s | strength, A $\rightarrow$ I | $g_E$ | 1 Hz | gain, E |
| $g_I$ | 4 Hz | gain, I | $g_A$ | 1 Hz | gain, A |
| $\sigma$ | $3.5\sqrt{2}$ | noise std | $\theta_E$ | $\in [-10, 20]$ | threshold, E |
| $\beta$ | $\in [0, 10]$s | strength, adaptation | $J_{AE}$ | 0.5 s | strength, E $\rightarrow$ A |
| $J_{AI}$ | 0.5 s | strength, I $\rightarrow$ A | | | |

and similarly, we model gliotransmitter release from the astrocytes as:

$$\tau_A \frac{dG_i^A}{dt} = -(G_i^A - G_L^A) + I_{rec,i}^A(t) + I_{ext,i}^A(t) \tag{9}$$

with $i \in \{1, \ldots, N_X\}$. $G_i^A$ is thus a phenomenological dimensionless variable that integrates the neuronal and astrocytic inputs to astrocyte $i$. According to the integrate-and-fire principle, whenever the membrane voltage of a neuron of population X exceeds its threshold $\theta_X$ at time $t$, a spike is emitted and the membrane voltage is reset to $V_r^X$. Similarly, when $G_i^A$ exceeds the threshold $G_{th}$, astrocyte $i$ emits a gliotransmitter release event, and $G_i^A$ resets to $G_r$. Gliotransmitter release events and spikes are then integrated in the corresponding synaptic variable $s_X$ (see Eqs (11) and (12) below).

For the simulations of the spiking model, we assumed the following connectivity rules:

- full connectivity for neuron-to-neuron connections and for astrocyte-to-astrocyte connections. The latter emulates the organization of astrocytes as a syncytium. This biological concept corresponds to the idea that all astrocytes of a local region are somehow interconnected together into single functional network [25].

- only a fraction (10%) of the E or I neurons are subjected to gliotransmission from the astrocytes. These neurons are chosen at random (uniform distribution).

- 50% of the astrocytes, chosen uniformly at random, receive inputs from the E or I neurons.

The latter two connectivity rules account from the observation that only part of the synapses of a given brain region are tripartite synapses contacted by astrocytes. The exact fraction seems to be quite variable from one region to the other, from 10% to 90% [25]. Therefore, our choice corresponds to a lower range of parameters.

One specificity of the original spiking model of [20] is to account for synaptic variables by a single pair of variables $u_X$, $s_X$ for each population, which can thus be considered as population variables instead of individual cell variables. Here we follow this model and define the recurrent input to each population $X = \{E, I, A\}$ as a population-level input as:

$$I_{rec,i}^X(t) = C_i^{XE} J_{XE} s_E(t) + C_i^{XI} J_{XI} s_I(t) + C_i^{XA} J_{XA} s_A(t) \tag{10}$$

where the A $\rightarrow$ E connectivity $C_i^{EA} = 1$ for 10% of the E neurons $i$ (chosen uniformly at random) and 0 for the others (the same for $C_i^{IA}$). For the (E, I) $\rightarrow$ A connectivity, we used $C_i^{AE} =$

**Table 2. Parameters used for the spiking model Eqs (7) to (13).**

| Parameter | Value | Definition | Parameter | Value | Definition |
|---|---|---|---|---|---|
| $\tau_E$ | 20 ms | time const., E | $\tau_I$ | 10 ms | time const., I |
| $\tau_A$ | 160 ms | time const., A | $\tau_a$ | 500 ms | time const., adaptation |
| $\tau_{\underline{E}}$ | 1 ms | time const., $u_E$ | $\tau_{\underline{I}}$ | 1 ms | time const., $u_I$ |
| $\tau_{\underline{A}}$ | 1 ms | time const., $u_A$ | $J_{EE}$ | 1.4 mV | strength, E $\rightarrow$ E |
| $J_{EI}$ | -1.4 mV | strength, I $\rightarrow$ E | $J_{II}$ | -1 mV | strength, I $\rightarrow$ I |
| $J_{IE}$ | 1.25 mV | strength, E $\rightarrow$ I | $J_{AA}$ | 0.16 | strength, A $\rightarrow$ A |
| $J_{AE}$ | 0.053 | strength, E $\rightarrow$ A | $J_{EA}$ | 22 mV | strength, A $\rightarrow$ E |
| $J_{IA}$ | 4.4 mV | strength, A $\rightarrow$ I | $J_{AI}$ | 0.058 | strength, I $\rightarrow$ A |
| $\beta$ | 1 ms | time const., adaptation | $K_a$ | 600 | strength, adaptation |
| $\sigma_E$ | 3 mV | noise std, E | $\sigma_I$ | 3 mV | noise std, I |
| $\sigma_A$ | 3 | noise std, A | $V_r$ | 14 mV | reset membr. pot. |
| $G_r$ | 9 | reset gliotrans. release | $V_{th}$ | 20 mV | spike-threshold |
| $G_{th}$ | 13 | gliotrans. release thresh. | $\tau_d^E$ | 23 ms | decay time, E |
| $\tau_d^I$ | 1 ms | decay time I | $\tau_d^A$ | 2 ms | decay time, A |
| $\tau_r^E$ | 8 ms | rise time, E | $\tau_r^I$ | 1 ms | rise time, I |
| $\tau_r^A$ | 8 ms | rise time, A | $d_{min}^E$ | 0 ms | min. delay, E |
| $d_{min}^I$ | 0 ms | min. delay, I | $d_{max}^E$ | 1 ms | max. delay, E |
| $d_{max}^I$ | 0.5 ms | max. delay, I | $d_{min}^A$ | 500 ms | min. delay, A |
| $d_{max}^A$ | 1.5 s | max. delay, A | $V_L^E$ | 7.6 mV | leak potential, E |
| $V_L^I$ | 6.5 mV | leak potential, I | $G_L^A$ | 7 | leak gliotrans. rate, A |

$C_i^{AI} = 1$ for 50% of the astrocytes (chosen uniformly at random) and 0 for the others. All the others connectivities ($C_i^{EI}$, $C_i^{EE}$, $C_i^{IE}$, $C_i^{II}$, $C_i^{AA}$) were set to 1 (all-to-all connectivity).

The synaptic variables $s_X$ integrate the spikes or release events emitted by all the cells in population X:

$$\tau_r^X \frac{du_X}{dt} = -u_X + \tau_X \sum_k \sum_{j=1}^{N_X} \delta(t - t_j^k - d_j^k) \tag{11}$$

$$\tau_d^X \frac{ds_X}{dt} = -s_X + u_X \tag{12}$$

with $t_j^k$ the $k^{\text{th}}$ spike (or release) time of cell $j$ of population X, $d_j^k$ its transmission delay (uniformly distributed between $d_{min}^X$ and $d_{max}^X$), and $\tau_r^X$ and $\tau_d^X$ the rise and decay times of the synapse, respectively.

Note that signal transmission in astrocytes is much slower than in neurons since it is based on reaction-diffusion (calcium signalling) instead of the propagation of an action potential [25, 37, 39]. To account for this important difference in timescales, we used transmission delays that were on the order of milliseconds for neurons ([0, 1] ms) but on the order of seconds for astrocytes ([0.5, 1.5] s, see Table 2).

In addition, the excitatory neurons displayed an after hyperpolarization (AHP) current:

$$\tau_a \frac{dI_{a,i}}{dt} = -I_{a,i} + \beta \sum_k \delta(t - t_i^k) \tag{13}$$

The external input current $I_{ext,i}^X(t) = \sigma^X \sqrt{\tau_X} \eta_i(t)$ is a Gaussian white noise term.

Initial conditions were set as $V_i^X = V_r + (V_{th} - V_r)\eta_i$ and $G_i^A = G_r + (G_{th} - G_r)\eta_i$, where $\eta_i$ is a random value with uniform distribution between 0 and 1. Unless indicated, we simulated the spiking network model using $N_E = 4,000$ excitatory neurons, $N_I = 1,000$ inhibitory neurons and $N_A = 2,000$ astrocytes. Each of these 2,000 astrocytes thus impacts 400 excitatory and 100 inhibitory neurons by gliotransmitter release, whereas half of them are individually impacted by the activity of the totality of the 4,000 E and 1,000 I neurons.

The values of the parameters in the equations above are given in Table 2. A fixed-point and linear stability analysis of the spiking model defined by Eqs (7) to (13) is provided in S2 Text.

## Parameter estimation

The experimental data available for parameter estimation of the two models above exhibit strong variability. In [20], for instance, the numbers used to quantify the experimental measurements (distribution of duration of Up or Down phases, CV of the firing rates) vary significantly from one repetition of the experiment to the other. Over the seven repetitions of the same experiments (their Fig 2A and 2B), the mean duration of the Up phases varies from 0.24s to 0.73s, for example. With other experimentalists, using other experimental setups, and recording on different cortical regions, the variability would probably be even larger. Therefore, the classical methodologies for parameter estimation would at best allow to match one specific repetition of a given experiment, where our objective here is to get a more generic overview of this system. Therefore, we have opted for an ad hoc method to set the values of the parameters. When available, we have set the initial guess for the parameters to rough estimates from the literature. For instance, the quantification of propagation delays of calcium waves in 3D astrocytes made in [39] sets an order of magnitude of around 1 to 2 sec for the maximal delay of astrocytes $d_{max}^A$. For the other parameters, we used bifurcation studies like those shown below to locate regions of the parameter space in which model simulations are approximately in agreement with the variation range of the main experimental quantities of [20]. The results are given in Tables 1 and 2.

## Automatic segmentation of Up and Down phases

We quantified the statistics of Up and Down phases in rate model or spiking network simulations on the basis of the mean firing rate time series. In spiking network simulations, we first computed the mean population rate from the raster plot, using a sliding window of 10 ms and counting the total number of spikes emitted by all neurons (excitatory and inhibitory) in the window. Automatic segmentation of the firing rate time series into Up phases and Down phases was achieved by smoothing the sampling rate using a sliding window of +/− 50 points around each data point and replacing each data point by the median over the window. Transition of the smoothed data curve through a threshold of 1.0 Hz from below was considered a switch from a Down to a Up state, whereas transition from above signaled a reverse switch, from Up to Down state. The first and last phases of a simulation were systematically discarded and not accounted for in the statistics.

## Code availability

The code used to generate the results of this article is freely available online on ModelDB: http://modeldb.yale.edu/267310. The scripts to simulate the rate model are rate_simu.ipynb (simulations of the model) and rate_diagram.ipynb (to find the fixed points). Likewise, the scripts for the spiking model are IAF.ipynb (simulations of the model), fixed_points_with_astro.ipynb (self-consistent equations to find fixed points, with astrocytes), fixed_points_woastro.ipynb (self-consistent equations to find fixed points, without astrocytes), stability.ipynb

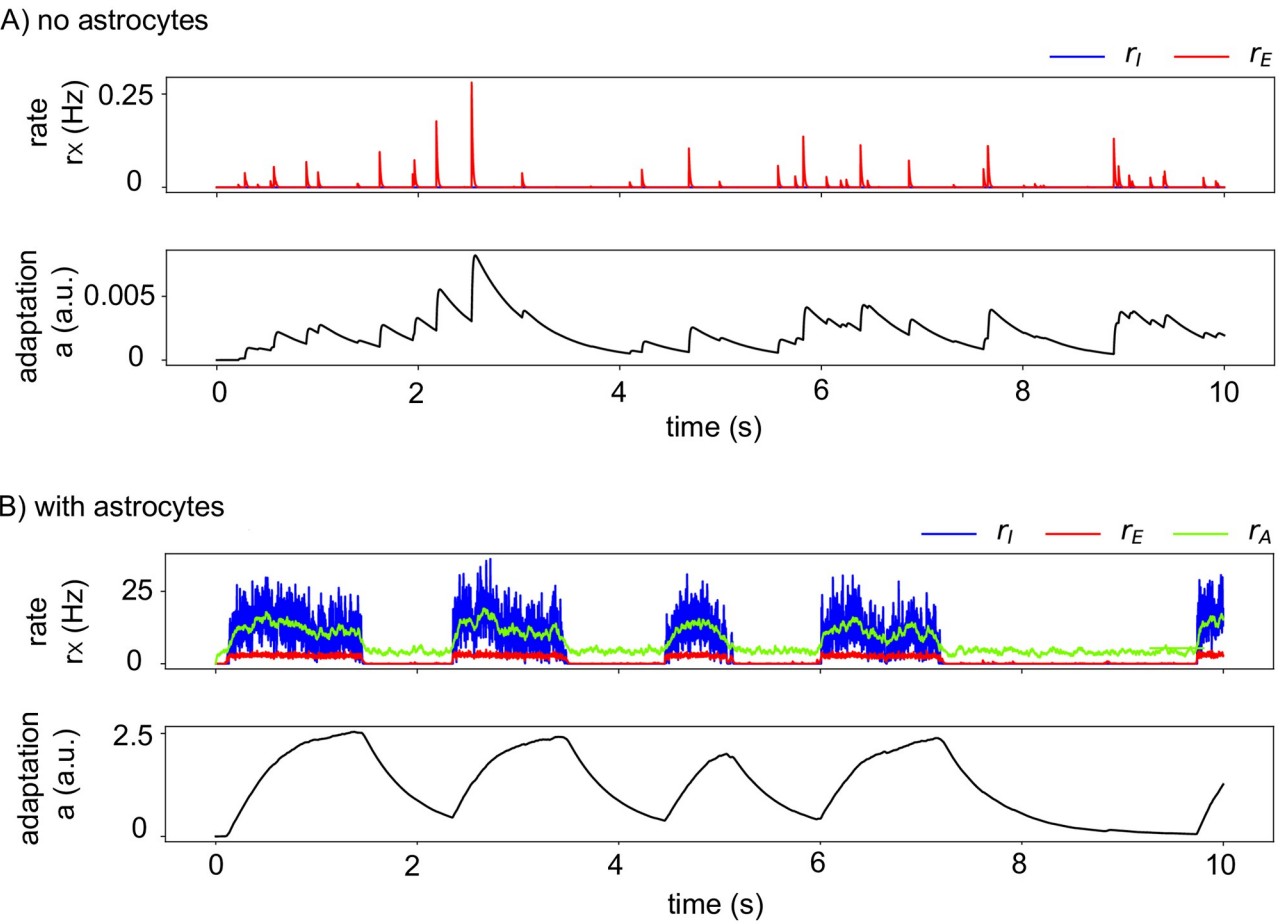

**Fig 2. Astrocytes in the rate model switch the dynamics from silent to Up-Down.** (See Section Materials and methods for the Rate model) (A) In the absence of astrocytic impact on the neurons ($J_{EA} = J_{AE} = J_{IA} = J_{AI} = 0$ s) the neurons are in a silent state with vanishing firing rates $r_E$ (*red*) and $r_I$ (*blue*) and adaptation $a$ (*black*), corresponding to a Down-state fixed point. Please note the difference of $y$-scale between panels (A) and (B). (B) When gliotransmission between astrocytes and neurons is accounted for ($J_{EA} = 1$, $J_{IA} = J_{AI} = J_{AE} = 0.5$ s), *with no change of the other parameters*, the dynamics switches to Up-Down dynamics. All other parameters given in Table 1.

(stability analysis of the fixed points), as well as UD_analysis.m (analysis of the Up and Down states durations), and FP_analysis.m (3d nullclines to find fixed points, with astrocytes).

## Results

### Rate model

We first illustrate the dynamics of the rate model described in Section Materials and methods for the Rate model by the simple numerical simulations of Fig 2. In the absence of astrocytes (i.e. with $J_{IA} = J_{EA} = J_{AI} = J_{AE} = 0$ s, Fig 2A), the model with the parameters of the figure is silent: the firing rate of the inhibitory neurons $r_I$ vanishes, and that of the excitatory neurons, $r_E$, is also zero most of the time, except for small fluctuations due to external noise. Accordingly, adaptation is essentially off. We then added gliotransmission between excitatory neurons and astrocytes in Fig 2B keeping all other parameters identical to Fig 2A. Adding gliotransmission drastically changes the dynamics that now exhibits spontaneous transitions between long periods of silence for all neuronal populations and shorter periods of high firing rates for the

excitatory and inhibitory neurons (around 10 and 5 Hz, respectively). In other words, astrocyte activity in the rate model switches the dynamics from silent to a Up-Down oscillatory dynamics, in agreement with experimental observations *in vivo* [37]. During an Up state, adaptation slowly increases and eventually triggers the Up-to-Down transition that ends the Up state.

Note that the average values of $r_E$, $r_I$ and $r_A$ during the Up and Down states in the simulation of the figure match the values predicted by the stability analysis of S1 Text. In particular, this analysis states that in the Down fixed-point, one expects $r_A = -g_A\theta_A/(1 - g_AJ_{AA})$ while the neuronal rates vanish. In agreement, the rate of gliotransmitter release by the astrocytes $r_A$ remains elevated during the Down states of Fig 2B, even though the neurons are silent.

To analyze further these simulation results, Fig 3 summarizes the fixed-point and linear stability analysis of S1 Text in the $(\beta, \theta_E)$-parameter plan. In the absence of noise, Fig 3A, the system behavior is determined by two straight lines: the Down steady-state exists (and is stable) only on the right hand side of the line defined by Eq (S1.5) in S1 Text whereas the Up steady-state exists for the half-plan below the line defined by Eq (S1.14) in S1 Text. This defines two regions of mono-stability, one where the Up state is the only fixed-point (U-region) and the other where the Down state in the only one (D-region). The region where both the Up state and the Down state exist is a region of bistability ("Bist." in the figure) where the dynamics converges to the Up or the Down state depending on the initial conditions. Finally, in the region where neither the Up nor the Down fixed-points exist, the arguments of the rectification functions regularly switch from positive to negative and back. This yields a regime of oscillations ("Osc"-region), that is a specific manifestation of the non-smooth character of the model.

With noise (Fig 3B and 3C), spontaneous Up-Down transitions are expected to occur in the bistable and oscillatory regions of the noiseless system, but also in sub-regions of its U- and D-regions (see S1 Text). Altogether, this defines three dynamical regimes: low values of $\beta$ and $\theta_E$ are predicted to give rise to a regime of perpetual high firing rates, i.e. a stable Up state. Conversely, large values of the excitatory neuron threshold $\theta_E$ are expected to yield a silent regime, or Down state, where neuronal firing rates vanish. Between those two regions, the system is predicted to switch spontaneously between periods of high population rates and periods or collective silence, i.e. $U \leftrightarrow D$ dynamics. A theoretical estimation for the frontier between the $U$ and the $U \leftrightarrow D$ region with noise is given by Eq (S1.16) in S1 Text (Fig 3, *dashed black lines*). Comparing with the percentage of the simulation-time spent in the Up state shows that in all the cases illustrated in Fig 3B and 3C this expression indeed correctly positions the *D*-region on the plan, although it strongly underestimates its size. Likewise Eq (S1.15) in S1 Text (Fig 3, *dashed green lines*) indeed indicates the transition between the *D* and the $U \leftrightarrow D$ regions, although, here again, the predicted size of the *D* region is strongly underestimated.

Remarkably, this frontier between $U \leftrightarrow D$ and $D$ is very sensitive to modifications of gliotransmission couplings (the $J_{XA}$s and $J_{AX}$s): the presence of astrocytes indeed pushes this frontier to larger values of $\theta_E$. The cyan star in Fig 3B and 3C locates the parameter values used in Fig 2. Without astrocytes, the star is located on the right hand side of the frontier between the $U \leftrightarrow D$ and the $D$ region, thus explaining the silent state of Fig 2A. With the addition of gliotransmission, the frontier moves rightwards, so that now, the star is located inside the $U \leftrightarrow D$ region. This explains the Up-Down regimes of Fig 2B and 2C.

Taken together, these results indicate that astrocyte activity can indeed switch the network dynamics from silence to the Up-Down regime by altering the phase diagram of the dynamics. The effect of gliotransmission on the model dynamics is not drastic, in particular gliotransmission does not change the nature or the number of bifurcation points in the system. However, it displaces the frontiers separating dynamical regimes, thus allowing the expression of Up-Down dynamics for a larger range of values of the firing threshold of the excitatory neurons.

 

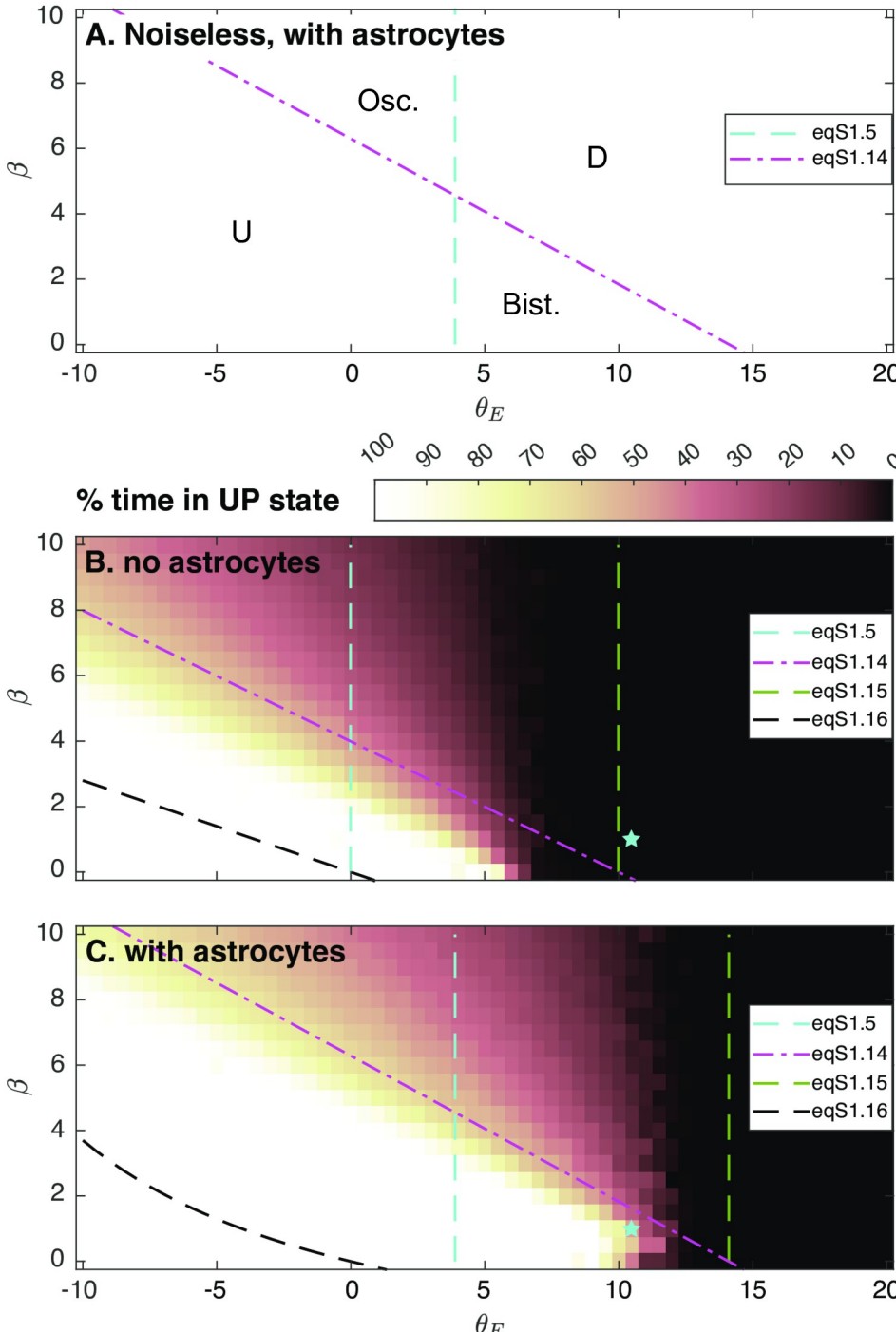

**Fig 3. Prediction of the dynamical regimes of the rate model.** Diagram as a function of the adaptation strength $\beta$ and the threshold of the excitatory neurons $\theta_E$. (A) In the absence of noise, the deterministic model exhibits two regions of monostability: the "U"-region where the Up state fixed-point is the only one, and the "D"-region where the Down state fixed-point is the unique fixed-point. Both fixed-points coexist in the bistable region "Bist." whereas the dynamics oscillates in the "Osc." region. Those regions are precisely delimited by Eq (S1.5) and Eq (S1.14) in S1 Text. (A-B) With noise, three main regimes are predicted: a purely Up state in the bottom left part of the plan, a purely Down state in the right part of the plan and spontaneous transitions between Up and Down states in-between ($U \leftrightarrow D$). The color-code indicates the percentage of time spent in the Up state during a simulation. (A) Simulations were carried out with astrocytes ($J_{EA} = 1$, $J_{IA} = J_{AI} = J_{AE} = 0.5$ s, or (B) in their absence ($J_{IA} = J_{AI} = J_{EA} = J_{AE} = 0$ s. The cyan star locates the parameters of Fig 2, which shows in particular that gliotransmission pushes the frontiers of the Up-Down region

further to the right, effectively switching the dynamics to the Up-Down regime. Equation (S1.15) and Eq (S1.16) in S1 Text are theoretical estimates of the frontiers between $U \leftrightarrow D$ and $D$ or $U \leftrightarrow D$ and $U$. All other parameters given in Table 1.

### Stochastic spiking network model

To assess the above mechanisms in a more biophysically realistic circuit, we next expressed the circuit of Fig 1 as a stochastic spiking network model, with leaky integrate-and-fire neurons and astrocytes (Section Materials and methods for the Spiking model). Using the same illustration as for the firing rate model above, we start in Fig 4 with a network devoid of astrocytes, i.e. for which $J_{AE} = J_{AI} = 0$ and $J_{EA} = J_{IA} = 0$ mV. The neurons exhibit a very short firing phase at the beginning of the stimulation due to our choice of random initial conditions but quickly converges back to a silent state.

Adding gliotransmission between astrocytes and neurons strongly affects the dynamics (Fig 5): periods of nearly complete neuronal silence now spontaneously switch to periods of high collective neuronal firing, during which roughly all neurons fire on the order of 2 to 15 spikes. The raster plot of Fig 5B also suggests the factors that trigger Down-to-Up transitions: Up states are systematically initiated by a strong firing activity in the subset of excitatory neurons that are contacted by the astrocytes (neurons numbers 50 to 70 in the raster plot). This first wave of excitation then is transmitted to the whole populations of neurons (E and I), thus forming an Up state.

Hence, our biophysical model of stochastic spiking neurons confirms that astrocyte activity can switch the neuronal network from silent to Up-Down dynamics. During the Up states, the mean population firing rate of the spiking network is similar to te one exhibited by the firing rate model, i.e. around 10 Hz for inhibitory neurons and 5 Hz for excitatory ones (compare Figs 5C with 2B), confirming the good match between the rate and spiking models despite the dissimilarity of their spatiotemporal scales. The distributions of the duration of the Up and Down states are estimated in Fig 5D. For Down states the distributions is peaked around 0.5 seconds whereas it is much broader for Up states, with a large part of the durations comprised between 0.5 and 1.3 seconds. On average, the Down states are twice shorter than the Up states: $459 \pm 336$ ms for the Down states *versus* $1,031 \pm 575$ ms for the Up.

However, unlike the neurons that collectively synchronize their firing as successive Up and Down phases, the rate of gliotransmission events by astrocytes does not exhibit strong evidence of alternation between distinct activity phases (Fig 5, *green*). The membrane potential of the individual neurons is strongly bimodal, fluctuating around a lower mean value during Down phases and around a larger mean during Up phases, on top of which spikes are emitted (Fig 5A, *blue, red*). In strong contrast, the dynamics of the gliotransmitter release variables $G_i^A$ is devoid of such alternations, rather appearing to fluctuate around a single, stationary mean (Fig 5A, *green*). This opposition is also visible in the raster plot (Fig 5B): the neuronal spikes are strongly synchronized and their presence almost totally restricted to the Up phases, whereas the astrocytic gliotransmitter release events are emitted at an intermediate frequency, but with no clear variation of frequency between Up and Down phases. The evolution of the population synaptic variables, the $s_X$s of Eqs (10) to (12), provides another evidence that the neuronal and astrocytic dynamics are different (Fig 5C): the astrocytic variable $s_A$ (*green*) fluctuates around a low but constant mean, independently of the Up and Down phases of the neurons that strongly condition the neuronal synaptic variables $s_E$ (*red*) and $s_I$ (*blue*). Of course, the reason why the astrocytic release events are hardly synchronized along the Up and Down phases contrarily to the spiking activity of the neuronal populations is the difference of

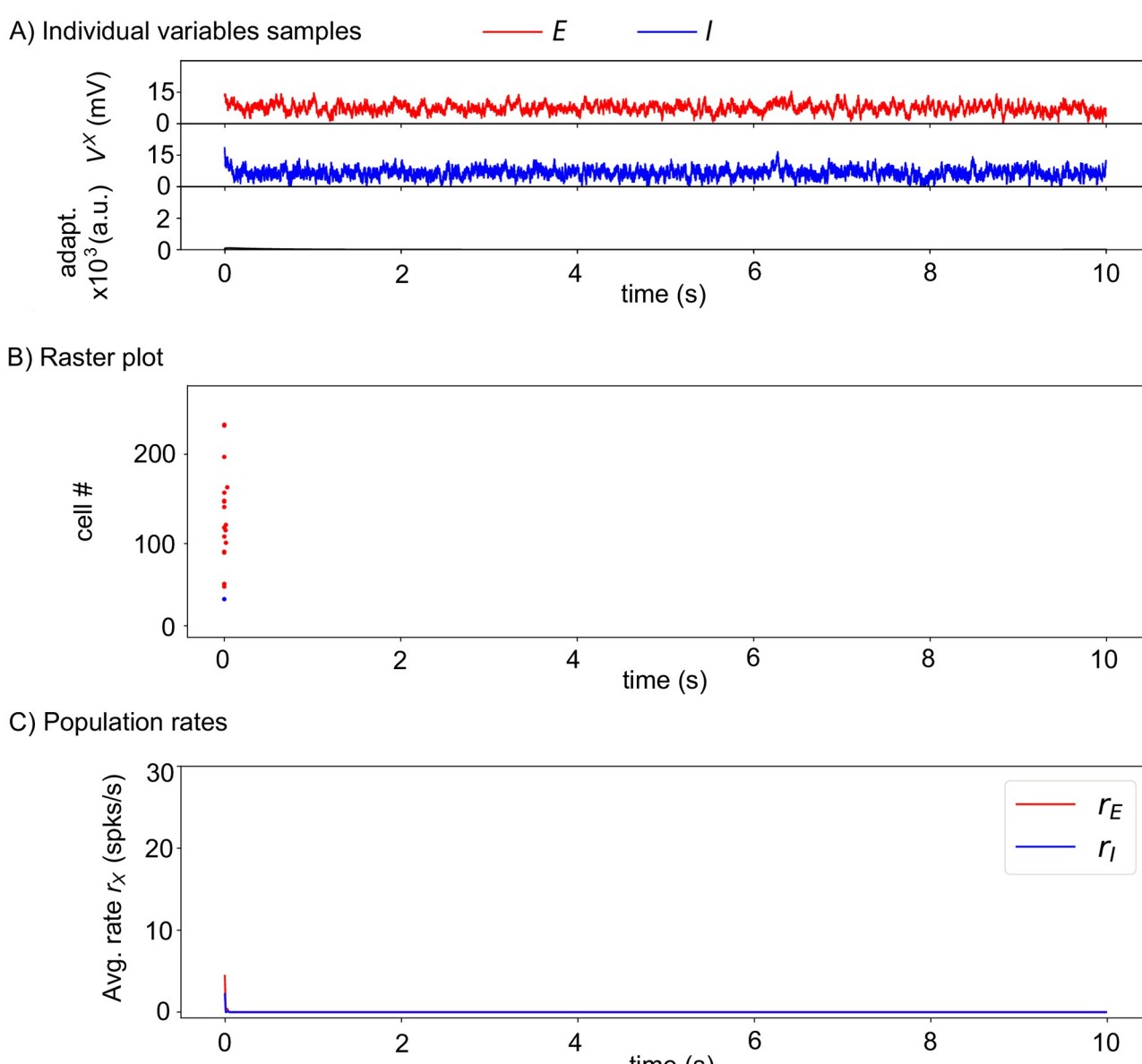

**Fig 4. Without astrocytes the stochastic spiking network is in a silent state.** (See Section Materials and methods for the Spiking model). Without astrocytes $J_{AE} = J_{AI} = 0$ mV, $J_{IA} = J_{EA} = 0$ mV. (A) Membrane voltages of two randomly chosen cells, one excitatory (*red*) neuron, one inhibitory (*blue*) neuron, as well as the average AHP current (*black*). (B) The spike rastergram that locates with points the spike times of a randomly-chosen subset of the neurons (one neuron = one row), and (C) the corresponding mean population rates are shown using the same color-code. The short initial burst of activity is due to the initial conditions where every cell is initiated randomly between its resting potential and the spiking threshold. $N_E = 4,000$ excitatory neurons, $N_I = 1000$ inhibitory neurons. Other parameters given in Table 2.

timescales for information transmission in those cells: on the order of milliseconds for neurons versus seconds for astrocytes (the $d_{min}^X$ and $d_{max}^X$ of Table 2).

Therefore, in the simulations of Fig 5, astrocytes provide the neurons with a constant, basal level of gliotransmission events that fuels their spontaneous collective alternation between Up and Down firing phases. Nevertheless, this background stochastic level of astrocytic input to the neurons is more than an additional random external input to the neurons. To show this, we went back to the spiking model without astrocytes of Fig 4 and increased the random

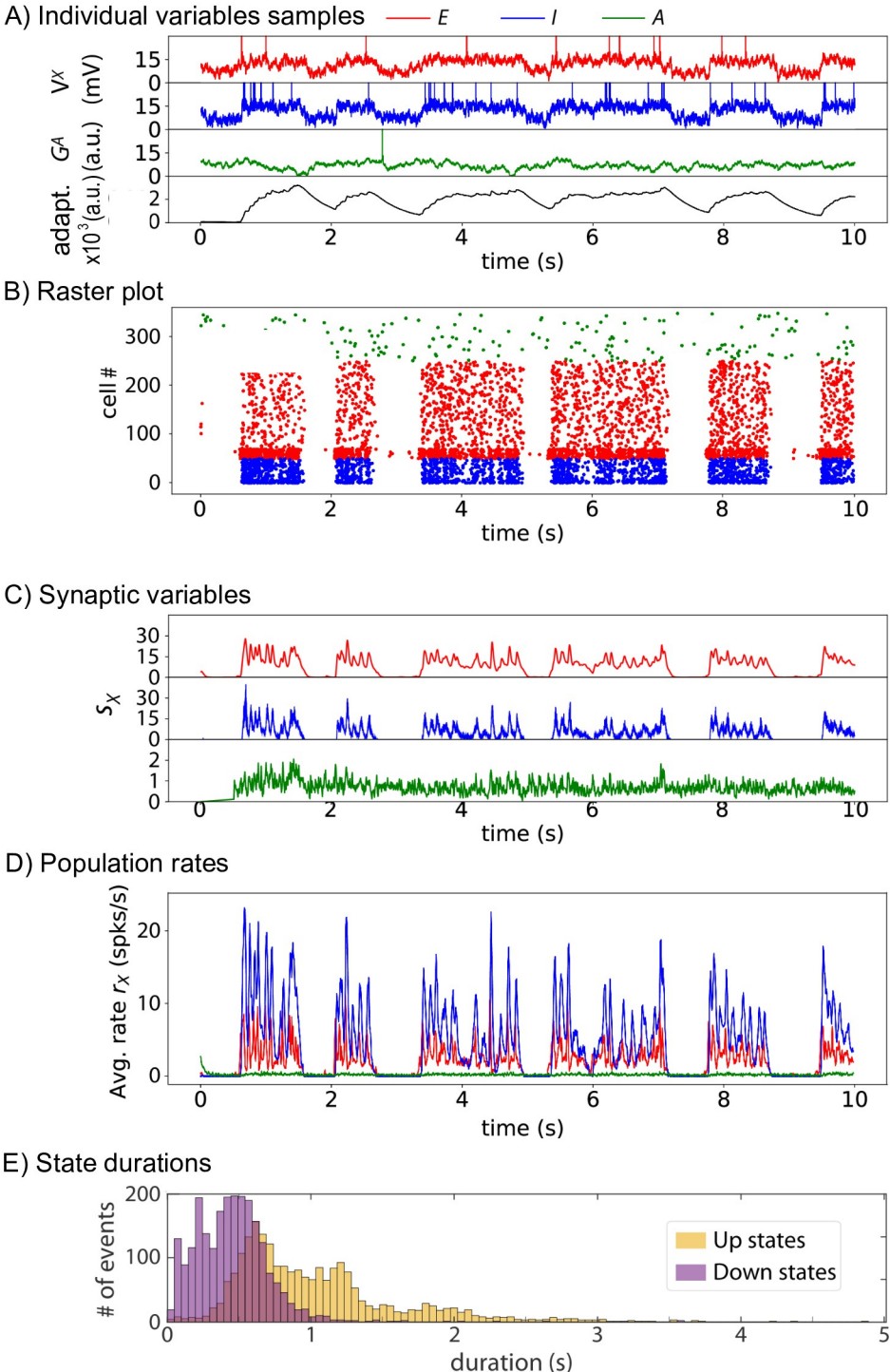

**Fig 5. With astrocytes, the stochastic spiking network switches to a Up-Down dynamic regime.** (See Section Materials and methods for the Spiking model). With astrocytes $J_{AE} > 0$ mV, $J_{AI} > 0$ mV. (A) Membrane voltages of three randomly chosen cells, one excitatory (*red*) neuron, one inhibitory (*blue*) neuron and the astrocyte (*green*) as well as the average AHP current (*black*). (B) The spike rastergram, (C) corresponding synaptic variables $s_X$ and (D) mean population rates are shown with the same color-code. (E) Distribution of Up (orange) and Down (purple) state durations for E and I cells (based on 200 independent simulations of 20 sec each, resulting in a total of 2273 Up states and 2356 Down). For each simulation, $N_E = 4000$ excitatory neurons, $N_I = 1000$ inhibitory neurons and $N_A = 2000$ astrocytes. Other parameters given in Table 2. For readability, the first phase of the simulation, characterized by a short very active up state, was discarded.

external input to the neurons. Fig 4 showed that the network is silent with the default value of the standard deviation of the random external input noise, $\sigma_X = 3$ mV (Table 2). Increasing $\sigma_X$ to 5 mV does give rise to an Up-Down regime with spontaneous alternation of Up and Down phases (Fig 6). However, the difference between the resulting Up and Down phases is much

### A) Individual variables samples

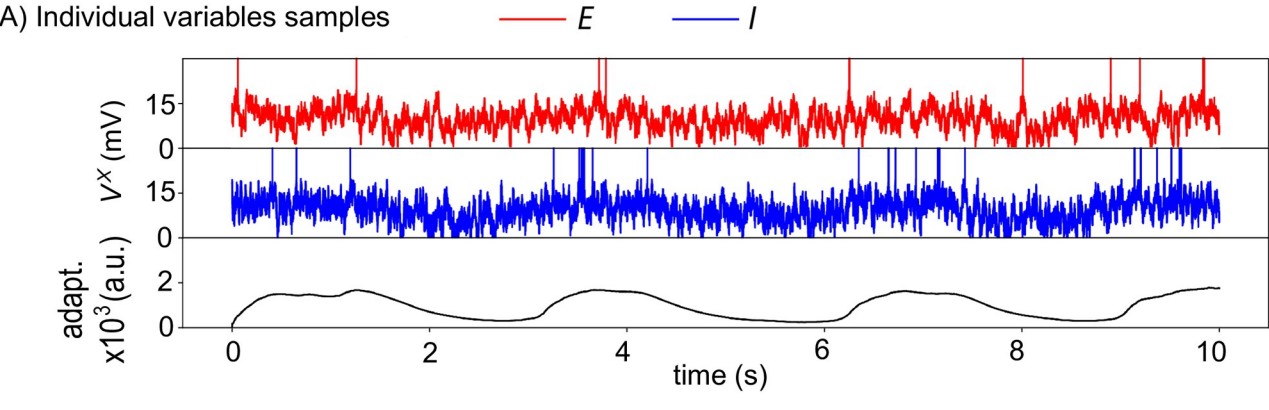

### B) Raster plot

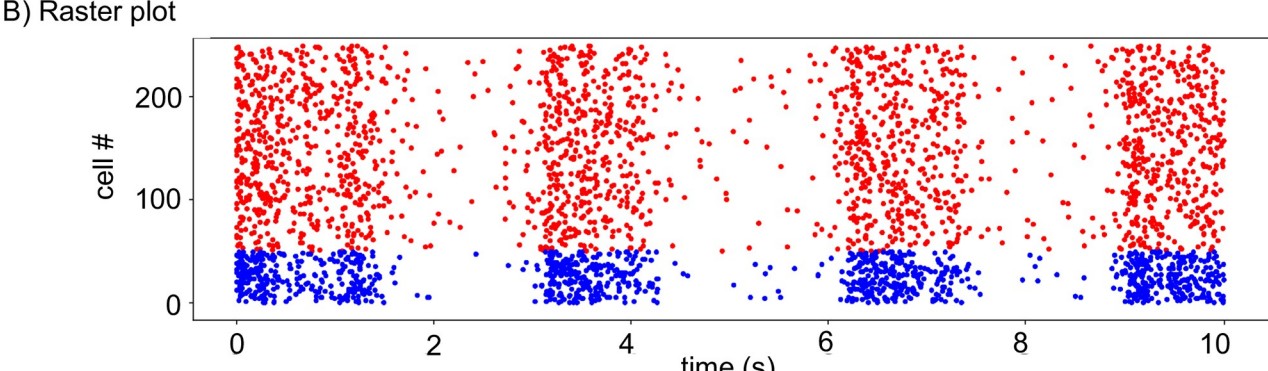

### C) Population rates

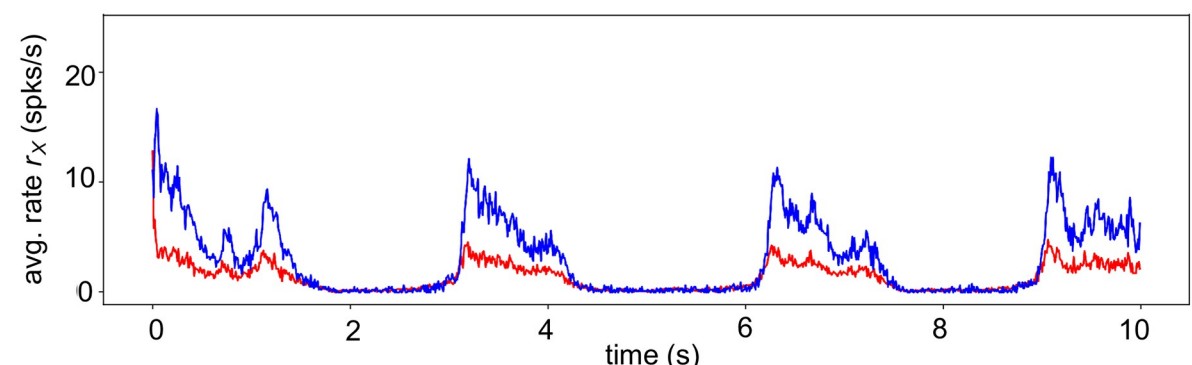

**Fig 6. Without astrocytes but with large external noise, the stochastic spiking network exhibits spontaneous Up-Down transitions.** (See Section Materials and methods for the Spiking model). Without astrocytes $J_{AE} = J_{AI} = 0$ mV, $J_{IA} = J_{EA} = 0$ mV. The amplitude of the stochastic external input is $\sigma_X = 5$ mV. However the difference between the phases is less marked than the dynamics observed with astrocytes. (A) Membrane voltages of two randomly chosen cells, one excitatory (*red*) neuron, one inhibitory (*blue*) neuron, as well as the average AHP current (*black*). (B) The spike rastergram and (C) the corresponding mean population rates are shown using the same color-code. All parameters are identical to those of Fig 4, except for the amplitude of the noise to the neurons $\sigma_E = \sigma_I = 5$ mV. $N_E = 4000$ excitatory neurons, $N_I = 1000$ inhibitory neurons.

less marked than in the Up-Down regimes with astrocytes: the subliminal individual membrane voltages are no more clearly bimodal (Fig 6A), and the difference between firing rates in Up and Down phases is much lower, with a significant firing activity during Down phases (Fig 6B and 6C).

Moreover, the range of external input amplitudes that give rise to Up-Down regimes without astrocytes is much more narrow than with astrocytes. Mean-field fixed-point and linear stability analysis of the stochastic spiking network model is shown in Fig 7 (see S2 Text for details). Two bifurcation diagrams are compared: in Fig 7A, astrocytes are absent, like in the simulations of Fig 4, whereas Fig 7B shows the same diagram when astrocytes are present, like in Fig 5. These bifurcation diagrams show the evolution of the fixed points and their stability when one varies the amplitude of the external noisy input, i.e. the standard deviation of the stochastic input to the E and I neurons, $\sigma_X$. Without astrocytes, the diagram shows a stable fixed-point corresponding to a low firing rate for low $\sigma_X$ values and a second stable fixed-point yielding a larger firing rate at large $\sigma_X$ values. In a narrow range of $\sigma_X$ values ([4.4, 4.5] mV), the two stable fixed points co-exist, together with a third intermediate unstable one (dashed line), thus evidencing a region of bistable dynamics (magnified in the inset). We also indicate with a gray-shaded region the parameter range where simulations of the spiking model evidence spontaneous transitions between Up and Down phases (like in Fig 6). The prediction of the mean-field analysis is not very precise regarding the location of the bistability region, which is probably a finite-size effect related to the finite number of neurons and astrocytes in the simulations. However, the theoretical analysis agrees very well with the narrowness of the bistability region observed in simulations, which confirms that noise-induced Up-Down regimes with the parameter values of Table 2 are observed only for a limited range of input intensities in the absence of astrocytes. In particular, this range is very far from the value $\sigma_X$ = 3 mV used in the previous simulations, explaining the silent state Fig 4.

Astrocytes modify the bifurcation diagram (Fig 7B): the values of the firing rates of the neurons in the Up and Down stable fixed-point are roughly the same as without astrocytes, but the range of $\sigma_X$ values for which bistability and Up-Down regimes are observed is incomparably larger, extending to much lower values. In particular, the bistability region now includes the value $\sigma_X$ = 3 mV, thus the Up-Down regime observed in Fig 4. Note also that the bifurcation analysis predicts that the rate of emission of gliotransmission events by the astrocytes should be very similar either in the Up or the Down state, in strong opposition to the neuronal firing rates. This explains the observation that the gliotransmission emission rate in Fig 5 did not vary much between Up and Down phases: all variables do follow the bistable dynamics of the whole system, but the branches of stable fixed-points for the astrocytes are much closer to each other than those for the neuronal firing rates.

Hence, as for the firing rate model studied in Section Results for the Rate model above, adding astrocytes in the spiking model does not drastically alter the nature or number of bifurcations, but relocates the bistability region in the parameter space so that a point in the parameter space that is out of the bistability region without astrocytes can find itself inside the bistability region by the addition of astrocytes, thus exhibiting Up-Down regime.

As a final remark, all the above results were obtained with $J_{AI} > 0$, i.e. a scenario where the firing activity of the inhibitory neurons directly increases the probability of gliotransmitter release by the astrocytes. However, we checked that the absence of this specific interaction does not jeopardize the validity of our conclusions here. We show in S1 Fig the results obtained with the rate model (S1A Fig) or the spiking network model (S1B Fig) when the strength of I → A interactions vanishes ($J_{AI} = 0$), while keeping all the other parameters as in Tables 1 or 2. This figure evidences a handful of changes compared to the scenario $J_{AI} > 0$ illustrated above, but the simulation results are still very similar (compare S1A Fig with Fig 3

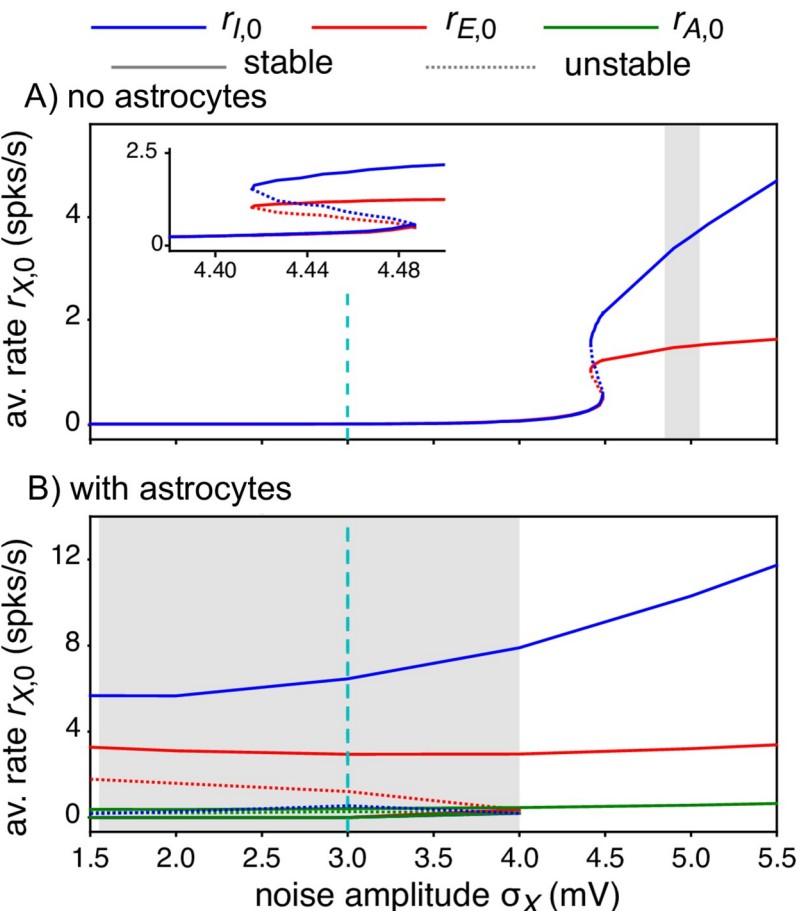

**Fig 7. Linear stability analysis of the spiking network model.** (A) Without or (B) with astrocytes along the intensity of the noisy external input to the neurons $\sigma_X$. In both cases, a bistable region is observed, ended by a saddle-node bifurcation for large $\sigma_X$. However, the bistable region is drastically reduced in the absence of astrocytes, as evidenced by the width of the gray-shaded region, that locates the range of $\sigma_X$ values for which Up-Down regimes are observed in numerical simulations of the network. These bifurcation diagrams show the evolution of the stable (solid lines) and unstable (dotted lines) fixed points of the equilibrium rates $r_{E,0}$ (*red*), $r_{I,0}$ (*blue*) and $r_{A,0}$ (*green*). In (A), the insets show a zoom out around the bistability region without astrocytes. See S2 Text for details on linear stability analysis. The dashed cyan vertical line indicates the $\beta$ value used for numerical simulations in Figs 4 and 5. Other parameters are given in Table 2. Note in particular that the diagram was obtained using $\sigma_E = \sigma_I \equiv \sigma_X$ and keeping a constant $\sigma_A = 3$.

with or S1B Fig with Fig 5), so that the conclusions drawn above are still valid in the absence of I → A interactions.

## Discussion

Up-Down cortical dynamics have primarily been observed during sleep or anesthesia. However, similar dynamical regimes have also been reported in the cortex during quiet wakefulness [40, 41] or during a task [42, 43]. Therefore making sense of these dynamics is important for our understanding of brain operations in general, not only during sleep or anesthesia. The cellular mechanisms that support the emergence of spontaneous Up to Down and Down to Up transitions in the cortex are however still unclear. The hypothesis that these transitions could be controlled by a mechanism intrinsic to the neurons of the considered cortical region has been explored by a number of theoretical or computational studies [8–11]. However recent

experimental studies reported the implication of other types of intrinsic brain cells, in particular astrocytes [36, 37, 44].

These results motivated us to propose our rate model Eqs (1) to (6). The main novelties here are *i*) to introduce the impacts of astrocytes in the dynamics of neuronal networks in the Up-Down regime and *ii*) to account for the influence of astrocytes using a rate equation with a similar mathematical structure as the firing rate equation of the neurons. Modelling the gliotransmitter release activity of astrocytes using a rate equation similar to the firing rate equation of the neurons enabled us to preserve the mathematical tractability of the model. We acknowledge that using Eqs (5) or (9) is a strongly simplified modeling of gliotransmission. However, it has the advantage of preserving the main biological ingredients of gliotransmission while keeping the model simple enough for the analytical study of its stability. We believe that the possibility to rely simulation results on an underlying sound theoretical analysis was important for the present article, and this is the reason why we have chosen to keep these population synapses.

In our numerical simulations, the addition of gliotransmission from astrocytes was sufficient to transform a neural network prepared in the Down, silent state into a dynamical regime of spontaneous alternations between Up and Down states. The inclusion of astrocytes in our model therefore provided us with the opportunity to explore how astrocytes alter the dynamics of the neuronal firing rates in a way that switches them to the Up-Down alternation regime. A major conclusion from our model is that gliotransmission probably does not have a drastic effect on the underlying dynamics of the network. Adding gliotransmission does not modify the number nor the type of the observed bifurcations, it only alters the values of the parameters at which these bifurcations occur. As a result, gliotransmission can transform a silent neural network model into a network exhibiting Up-Down dynamics, with no change of the neuron-related parameters, and no alterations of the neural mechanisms that control the transitions between Up and Down phases. Moreover, our model suggests that the fundamental differences of signal integration in neurons versus astrocytes may be crucial in the emergence of Up-Down regimes. In particular, the signaling delay in our spiking network model was kept three orders of magnitude larger in astrocytes compared to neurons, i.e. seconds versus milliseconds. This difference of timescales turned out to be crucial for the network dynamics illustrated in Fig 5 where a stationary background of astrocytic gliotransmission events triggers spontaneous transitions between synchronized Up and Down phases of neuronal firing.

The main limitation of our models are the simplification assumptions that we made to express the impact of astrocytes on the neural network model. The modeling literature proposes mathematical descriptions of the process of gliotransmitter release from astrocytes that are much more complex or accurate than the simple phenomenological expressions used here, see e.g., [38] for a recent account. However the price to pay for the added complexity would be a restriction of the available mathematical understanding of the system dynamics. Future numerical simulation works will be needed to assess whether the inclusion of such more complex descriptions comes with changes of the main conclusions of the present study. We also adopted the modelling choice made by [20] for their spiking model where the synapse dynamics are modelled using a single population variable, integrating the spikes emitted by the whole population into a single variable that can then be fedback to the other cells. This choice limits the range of modelling exploration regarding connectivity. It forbids models where the inputs received by an astrocyte is restricted to a subset of the neurons or, conversely, those where gliotransmission from an astrocyte targets only a subset of the synapses of a neuron. On the other hand, though, this modelling choice greatly facilitates theoretical (meawn-field) analysis of the stochastic network model. We leave for future works the study of models that would incorporate the main ingredients of our models above, but with real individual synapses and / or more

realistic sparse neuron-neuron connectivity [12]. This would make it possible to associate a spatial embedding to the network thus enabling the study of slow wave propagation [14] or to compare Up-Down regimes during sleep with those observed during anasthesia [13].

Experimental reports indicate that astrocytes form roughly 20 to 40% of all glial cells [25]. On the other hand, estimates of the ratio between glial cells and neurons in human cortex varies from 1.5 to more than 2 in humans [25]. Altogether, those numbers yield an astrocyte:neuron number ratio in the human cortex that ranges from 1:3 to 1:1. The numbers chosen for our simulations of the spiking network model are in good agreement with these experimental reports, with an astrocyte:neuron number ratio of 1:2.5. Additional comparisons can be made with the in vivo experiments reported in [20] from multichannel silicon microelectrode recordings in the somatosensory cortex of urethane-anesthetized rats. As explained in Section Materials and methods, Parameter estimation, we have set parameter values so that the model simulations exhibit behaviors similar to the experiments of [20]. We now give a more detailed account of the match between model and data. The distributions of Up or Down phase duration in the experiment shown in Fig 2A of [20] are broad, with Down phases lasting from less than 100 ms to 1.5 s and Up phases reaching larger maximal values, up to 2 s. Our simulation results exhibit similar broad distributions, at least for Up states, a consequence of the large variability of the Up state durations (Fig 5D). The coefficient of variations from the in vivo experiment of Fig 2A [20] were 0.61 and 0.70, for Up and Down phases, respectively, to be compared with 0.56 and 0.73 for our simulations. The mean values of the phase durations are also very well replicated by our simulations: 1.03 and 0.46 s for Up and Down phases, respectively, vs 0.65 and 0.38 in the experiment of [20], Fig 2A. The instantaneous population rate during Up phases in these in vivo experiments is around 4 to 6 Hz in [20] (their Fig 1C), a value that is similar to the population rate of excitatory neurons in our simulations (Fig 5D). Taken together, we thus conclude from those quantitative comparisons that our simulation results exhibit Up and Down phases that agree well with available experimental data.

The main experimentally-testable prediction made by our work is arguably the possibility of a dynamical regime where the astrocytic gliotransmitter release events are only weakly synchronized to the succession of Up and Down phases of the neuron firing state. In this regime the population frequency of gliotransmitter release events does not change much in Up phases compared to Down phases. Experimental testing of this prediction would consist in measuring simultaneously the activity of a local population of neurons using e.g., multi-channel silicon microelectrodes while monitoring the gliotransmitter events from astrocytes from the same local area. Gliotransmitter release events are difficult to monitor experimentally, even with glutamate-sensitive fluorescent reporters (see e.g., Fig 7D in [37]). Monitoring intracellular calcium activity could constitute a good proxy to locate glutamate release events by astrocytes. However, recent experimental studies have challenged the relation between calcium signals recorded from astrocyte cell bodies from those initiated in the fine processes, that are expected to contact the synapses [28, 29, 39]. Therefore, experimental testing of the above dynamical regime would probably need the measure of local calcium signals, within the fine astrocyte processes that form the so-called "gliapil". At any rate, this predicted dynamical regime is supported by activity-dependent release of gliotransmitters by astrocytes, which existence and impact on the neurons in physiological conditions is still debated among experimental neuroscientists [32, 33]. Therefore, according to the work presented here, experimental observation of astrocytes releasing gliotransmitters at a roughly constant rate while neurons undergo successive Up and Down firing phases, should be interpreted as an argument in favor of the existence of gliotransmission, and not against it.

## Supporting information

**S1 Text. Fixed points and linear stability analyses: Rate model.**
(PDF)

**S2 Text. Fixed points and linear stability analyses: Spiking model.**
(PDF)

**S1 Fig. Results in the absence of I → A interactions.** Results of the rate model of Eqs (1) to (6) (A) or the spiking network model of Eqs (7) to (13). (B) obtained in the presence of astrocytes, but with $J_{AI}$ = 0 mV, i.e. with no direct effect of inhibitory neurons on astrocytic gliotransmitter release. All parameters were as indicated in Tables 1 or 2, except for the value of $J_{AI}$ that was set to 0. Refer to Figs 3C and 5 for the color-codes and parameters of panels (A) and (B), respectively.
(TIF)

## Author Contributions

**Conceptualization:** Lisa Blum Moyse, Hugues Berry.

**Formal analysis:** Lisa Blum Moyse, Hugues Berry.

**Funding acquisition:** Hugues Berry.

**Investigation:** Lisa Blum Moyse, Hugues Berry.

**Methodology:** Lisa Blum Moyse, Hugues Berry.

**Project administration:** Hugues Berry.

**Resources:** Hugues Berry.

**Software:** Lisa Blum Moyse, Hugues Berry.

**Supervision:** Hugues Berry.

**Validation:** Hugues Berry.

**Visualization:** Lisa Blum Moyse.

**Writing – original draft:** Lisa Blum Moyse.

**Writing – review & editing:** Hugues Berry.

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
