## [Decision Letter · Decision Letter 0]

4 Apr 2022

Dear Dr. Berry,

Thank you very much for submitting your manuscript "Modelling the modulation of cortical Up-Down state switching by astrocytes" for consideration at PLOS Computational Biology. As with all papers reviewed by the journal, your manuscript was reviewed by members of the editorial board and by several independent reviewers. The reviewers appreciated the attention to an important topic. Based on the reviews, we are likely to accept this manuscript for publication, providing that you modify the manuscript according to the review recommendations.

Sincerely,

Michele Migliore

Associate Editor

PLOS Computational Biology

Daniele Marinazzo

Deputy Editor

PLOS Computational Biology

[LINK]

Reviewer's Responses to Questions

**Comments to the Authors:**

Reviewer #1: This work adds to previous theoretical studies investigating how Up-Down cortical dynamics is triggered by the interplay between an intrinsic activity-dependent negative feedback of the firing rate and an external input to the network. Here, the authors studied the modulation by astrocytes with mathematical models. Their work is based on an existing compartment model on Up-Down cortical dynamics, which was extended by the authors to study the effect of astrocytes. Both a simple rate and an otherwise equal stochastic modelling framework were set up. Necessary core information of the model and its parameters as well as assumptions are described in the main text and more details in the supplement, which also provides more insights to the applied fixed-point and linear stability analysis. Comparisons of structural model differences and analysis of the parameter space indicate that the presence of astrocytes can induce the emergence of Up-Down cortical dynamics and can provide more clear differences between firing rates in Up and Down phases than the model with no astrocytes. The results provide useful insights into the dynamic regime of the model, indicating that gliotransmission between astrocytes and neurons allows Up-Down cortical dynamics to occur under a broader range of conditions. Finally, the general results are further confirmed by robustness tests, e.g. relaxing the condition that J_AI > 0, which is good modelling practice.

Overall, the manuscript demonstrates sound modeling work, is well structured and is written in plain language understandable to nonspecialists. I am happy to report that I only have a few small comments.

Minor comments:

Figure 1: the latin “a” is shown instead of the greek letter alpha, which is referenced in the figure caption and model equations in the text.

Lines 457 to 466: Such a comparison to a specific empirical study is nice, but it should be clear whether some sort of optimization of parameter values has been used to get this agreement. Multi-objective optimization can be used to define a parameter set that is providing the best match. It would be interesting to see whether one, both or none of the models with and without astrocytes can be parameterized in a way providing a nearly perfect match to these empirical estimates.

Reviewer #2: In this paper, a theoretical framework is provided to test scenarios and hypotheses on the modulation of Up-Down dynamics by gliotransmission from astrocytes. Three populations’ cells are considered which are interconnected by gliotransmission events from neurons to astrocytes and back. Two models are derived for this three-population system: a rate model and a stochastic spiking neural network. Even the paper is well written and involves an interesting idea; however, it requires some corrections. The authors must arrange their paper once again using following comments:

1) Concise the abstract providing quantitative analysis as well as main features of the problem. Include other detail in discussion section.

2) Difficult wording is used throughout the manuscript. There is no need for ambiguous wordings. Just make it simple and readable.

3) Is there any base to assume that each population receives a fluctuating external input?

4) What is new in the model equations (1) to (4) and why these are considered? Discuss in the paper.

5) Correlate results of three-population systems with the earlier ones (even in limiting cases).

Reviewer #3: I read the manuscript entitled " Modelling the modulation of cortical Up-Down state switching by astrocytes" with great interest. It has been a comprehensive study, and I think it has valuable content. However, the following significant corrections seem necessary to improve the scientific level of the article.

1- Please rewrite the abstract section. It has a lot of unnecessary descriptions.

2- Please explain more clearly the limitation and future work.

3- The discussions should highlight why the proposed method is providing good results. The comparative study from the recently proposed approach is missing.

4- How did the authors assume the connectivity rules?

5- The "Discussion" section should be added in a more highlighting, argumentative way. Please note that the up-to-date references will contribute to the up-to-date of your manuscript.

6- Please add a description for each MATLAB/Python code. It might be better to clarify which code is used in each manuscript section.

7- The introduction section needs to highlight the motivational contribution of the research

Reviewer #4: Thank you for a well written manuscript with an excellent approach and design.

1.my major concern is the lack of discussion how the system adjust to the biology data. network synchronization has ben described for decades and now we have a better model, however, the discussion could be a lot better

2. How the parameter were selected for the analysis need a better rationale and expansion of this section

3. please check the numbers of inputs in page 13. I think the numbers are bigger

Great manuscript

**Have the authors made all data and (if applicable) computational code underlying the findings in their manuscript fully available?**

Reviewer #1: Yes

Reviewer #2: None

Reviewer #3: Yes

Reviewer #4: Yes

PLOS authors have the option to publish the peer review history of their article (what does this mean?). If published, this will include your full peer review and any attached files.

Reviewer #1: No

Reviewer #2: **Yes: **Agree

Reviewer #3: **Yes: **Amirmasoud Ahmadi

Reviewer #4: No

Figure Files:

Data Requirements:

Reproducibility:

References:

---

## [Editor Report · Decision Letter 1]

11 Jun 2022

Dear Dr. Berry,

We are pleased to inform you that your manuscript 'Modelling the modulation of cortical Up-Down state switching by astrocytes' has been provisionally accepted for publication in PLOS Computational Biology.

Best regards,

Michele Migliore

Associate Editor

PLOS Computational Biology

Daniele Marinazzo

Deputy Editor

PLOS Computational Biology

---

## [Editor Report · Acceptance letter]

18 Jul 2022

PCOMPBIOL-D-22-00403R1 

Modelling the modulation of cortical Up-Down state switching by astrocytes

Dear Dr Berry,

I am pleased to inform you that your manuscript has been formally accepted for publication in PLOS Computational Biology. Your manuscript is now with our production department and you will be notified of the publication date in due course.

With kind regards,

Zsofi Zombor
